# Phenotyping Zebrafish Mutant Models to Assess Candidate Genes Associated with Aortic Aneurysm

**DOI:** 10.3390/genes13010123

**Published:** 2022-01-10

**Authors:** Andrew Prendergast, Bulat A. Ziganshin, Dimitra Papanikolaou, Mohammad A. Zafar, Stefania Nicoli, Sandip Mukherjee, John A. Elefteriades

**Affiliations:** 1Yale Zebrafish Phenotyping Core, Yale University School of Medicine, New Haven, CT 06510, USA; andrew.predergast@yale.edu (A.P.); stefania.nicoli@yale.edu (S.N.); 2Aortic Institute at Yale-New Haven, Yale University School of Medicine, New Haven, CT 06510, USA; bulat.ziganshin@yale.edu (B.A.Z.); dimitra.papanikolaou@yale.edu (D.P.); mohamad.zafar@yale.edu (M.A.Z.); sandip.mukherjee@yale.edu (S.M.); 3Yale Cardiovascular Research Center, Cardiology, Internal Medicine and Genetics, Yale University School of Medicine, New Haven, CT 06510, USA

**Keywords:** variant of unknown significance, VUS, whole exome sequencing, thoracic aortic aneurysm, zebrafish, *EMILIN1*, *MIB1*

## Abstract

(1) Background: Whole Exome Sequencing of patients with thoracic aortic aneurysm often identifies “Variants of Uncertain Significance” (VUS), leading to uncertainty in clinical management. We assess a novel mechanism for potential routine assessment of these genes in TAA patients. Zebrafish are increasingly used as experimental models of disease. Advantages include low cost, rapid maturation, and physical transparency, permitting direct microscopic assessment. (2) Methods: Zebrafish loss of function mutations were generated using a CRISPRC/CAS9 approach for *EMILIN1* and *MIB1* genes similar to VUSs identified in clinical testing. Additionally, “positive control” mutants were constructed for known deleterious variants in *FBN1* (Marfan’s) and *COL1A2*, *COL5A1*, *COL5A2* (Ehlers-Danlos). Zebrafish embryos were followed to six days post-fertilization. Embryos were studied by brightfield and confocal microscopy to ascertain any vascular, cardiac, and skeletal abnormalities. (3) Results: A dramatic pattern of cardiac, cerebral, aortic, and skeletal abnormalities was identified for the known pathogenic *FBN1* and *COL1A2*, *COL5A1*, and *COL5A2* mutants, as well as for the *EMILIN1* and *MIB1* mutants of prior unknown significance. Visualized abnormalities included hemorrhage (peri-aortic and cranial), cardiomegaly, reduced diameter of the aorta and intersegmental vessels, lower aortic cell counts, and scoliosis (often extremely severe). (4) Conclusion: This pilot study suggests that candidate genes arising in clinical practice may be rapidly assessed via zebrafish mutants—thus permitting evidence-based decisions about pathogenicity. Thus, years-long delays to clinically demonstrate pathogenicity may be obviated. Zebrafish data would represent only one segment of analysis, which would also include frequency of the variant in the general population, in silico genetic analysis, and degree of preservation in phylogeny.

## 1. Introduction

Marfan syndrome was described in 1896. Ehlers–Danlos syndrome, although first observed by Hippocrates, was described in modern medicine in the early 1900s. Loeys–Dietz syndrome was first reported in 2005. These are “syndromic” manifestations of thoracic aortic aneurysm disease (TAA), in which other organ systems besides the aorta show clinically discernible features.

In 1997 and 1999, Milewicz and our Yale team independently described essentially identical family patterns of “non-syndromic” TAA, in which the aorta (with its central branches) was affected, without external stigmata of inherited disease [1,2].

In the interim, advances in molecular genetics, especially the advent of affordable Whole Exome Sequencing achieved in the years since the first publication of J. Craig Venter’s gene sequence by Celera in 2001, [3] have permitted light-speed progress in identifying new causes of thoracic aortic disease. The contributions of Milewicz and her team have been monumental. This has led to at least 34 specific genes currently implicated in the causation of TAA. New causative genes are added every year. The sum total of causative genes is published in regular review articles in the journal AORTA [4].

When a suspicious new variant at a specific allelic location is identified, it is classified as a “Variant of Uncertain Significance (VUS)”, based on the guidelines of the American College of Medical Genetics and Genomics [5,6]. The variant remains so classified until strong information accumulates of its being truly causative rather than an incidental finding. A natural friction is encountered between the geneticist (who wishes not to be proven wrong by prematurely “calling” a potentially innocuous gene) and the clinician (often a surgeon), who cannot wait 5, 10, or 15 years for more families to be uncovered to confirm unequivocally that the specific variant segregates with the disease phenotype (aneurysm or no aneurysm) in multiple family trees. The patient can easily be lost to aortic dissection while unassailable confirmation of causative status of a variant is pursued [7].

Unlike many diseases, which receive small genetic input from multiple variants in many genes, TAA is special. Most genetically triggered TAAs (that are known to date) are caused by a single nucleotide polymorphism (a variation in a single base in a DNA sequence) in a single gene [8,9].

There are several general methods to pursue initial analysis of the potential significance of a VUS:

Frequency in the general population. The rarity of the variant is important. For TAA, a rather unlikely disease, the variant has to be rare, certainly less than 1/10,000 human beings (since according to epidemiologic studies, TAA is estimated to occur in 10 out of 100,000 individuals in the general population per year). If the variant is not rarer than that level, it is not of causative interest (unless we suspect a recessive type of TAA, which is very rare), because the variant is more common than the disease itself. Large databases are now available that include maps of more than 140,000 individuals (such as the Genome Aggregation Database (gnomAD) [10]).

In silico analysis. Computerized methods can predict the molecular impact of a specific variant at a specific spot in one particular gene. Some variants may be of almost no impact. Others can stop cold the reading of the gene beyond the variant, with disastrous protein manufacturing consequences. A dramatic negative effect on reading of the gene suggests the variant in question is likely disease-causing.

Conservation in phylogeny. The degree of conservation of the normal allele at one specific location in one gene is evidence of its importance. The further down the phylogenetic tree that the conservation of the allele goes, the more important is that location on that gene. Computerized methods are available for exact determination of degree of conservation of the normal variant at any specific location (from human to monkey, pig, dog, rat, mouse, fish). Conservation of a specific variant in phylogeny is among the strongest predictors of its importance in causation of disease.

Animal models. It is possible to insert the variant allele into an animal embryo and monitor for disease development. This has generally been done in mammals (mice) and is very time-consuming and expensive [11].

Familial Variant Segregation Analysis. Other affected and unaffected members of the same family can be tested for the VUS carrier status in order to determine whether the VUS segregates with the disease or not. Of course, issues of (1) degree of penetrance (the likelihood that a carrier manifests the disease) and (2) duration of time after birth required for an aortic aneurysm to manifest (usually decades) need to be taken in to account when conducting familial segregation analysis.

In this report, we describe a novel approach for efficient clinical analysis of significance of TAA variants via inducing genetically mutant zebrafish. Experiments in zebrafish are generally inexpensive and highly reproducible and yield rapid results because of the quick maturation of the species. Furthermore, the translucent nature of the zebrafish renders gross and microscopic analysis of internal organs more facile than in other species.

We reasoned that if the zebrafish method was feasible and informative, this model could provide a means for rapid assessment of the significance of a mutation and permit rapid introduction of that mutation into clinical decision making. Even if the precise clinical mutations cannot be recreated in the zebrafish in a rapid fashion, the associated genes can be rapidly screened for potential significance in a matter of weeks or months by inactivating them and studying zebrafish larval phenotypes. If successful, this rapid approach could be of life-saving significance in the TAA population—via earlier recognition of disease causation, with consequently expedited treatment.

## 2. Materials and Methods

We examined the impact of inducing loss of function mutations in genes potentially implicated in thoracic aortic aneurysm disease on anatomy and clinical outcome in zebrafish. We also examined the impact on zebrafish of mutations in some genes known to be implicated in Marfan’s disease (*fbn1*) and Ehlers–Danlos syndrome (*col1a2*, *col5a1*, and *col5a2a/b*). We used these known disease related genes as “positive controls”, which should be reflected morphologically if the zebrafish model were effective as an evaluative tool.

General zebrafish maturation and care methodology. Adult AB zebrafish were maintained on a standard 14 h/10 h dark/light cycle in 3.0 L housing with constant water flow (Iwaki Aquatic, Holliston, MA, USA). Adults were fed a standard lab diet of Gemma Micro 500 and brine shrimp, once daily (Skretting, Stavanger, Norway). Zebrafish embryos were obtained by natural matings and maintained in E3 medium (5 mM NaCl, 160 µM KCl, 333 µM CaCl_2_, 416 µM MgCl_2_) at 28.5 °C. All anesthesia was carried out in 0.16 mg/mL MS-222 (Syndel, Ferndale, WA, USA). Care protocols were modeled after standard methods at Boston Children’s Hospital [12,13]. All animal care and handling were carried out under the approval of IACUC (Protocol #: 2019-20274).

Validation of CRISPR reagents and generation of zebrafish transient mutants. For each of the eight genes analyzed in this study, two sequence-specific crRNAs were designed using the CRISPOR web tool (Table 1) [14]. These were annealed to tracRNA to form an RNA duplex stock (33 µM) which was then complexed with 10 µg/µL Alt-R-*S.p.* Cas9 Nuclease V3 (IDT) in Cas9 buffer (20 mM HEPES, 150 mM KCl, pH 7.5) at a 1:1 molar ratio according to the manufacturer’s instructions. This was then supplemented with sterile 2% phenol red to a final concentration of 0.2% phenol red containing a roughly 3.3 µM concentration of each CRISPR/Cas9 ribonucleoprotein complex. This constituted the injection mixture.

Zebrafish embryos were obtained at the 1–2 cell stage and 1 nL of the described mixture was injected into each embryo. Embryos were allowed to recover in E3 medium. ~10 embryos were collected at 1–3 days post fertilization (dpf) and their DNA was harvested by boiling them in 50 µL of 100 mM NaOH for 20 min. The pH of the resulting solution was adjusted by adding 20 µL 1M Tris-HCl pH 7.5.

Teleosts, of which zebrafish are a member, underwent a whole genome duplication event approximately 350 million years ago [15]. Consequently, they often have redundant paralogues of mammalian genes which necessitates inactivating both copies. This was the case for both *emilin1* and *col5a2*, which each have two paralogues *a* and *b* in zebrafish. We developed two gRNAs for each paralogue and injected a mixture containing all four gRNAs with Cas9 in an attempt to exclude any compensatory function. This method has been successfully applied for other diseases, including phenotypic screening of genes involved in spinal cord injury or defective locomotor behaviors [16,17,18].

These DNA preparations were subjected to site-specific PCR (Table 1) to detect heteroduplex DNA generated by CRISPR/Cas9 indels. The same PCR amplicons were isolated by TOPO cloning and directly sequenced to assess the efficacy of each gRNA individually.

Briefly, a 300–800 bp amplicon was generated at each targeted locus in both targeted and non-targeted (i.e., unmodified) fish. These were then denatured and reannealed using an extended 0.2 °C/s ramp. The resulting DNA was digested with T7 endonuclease (NEB) and run out on 2% agarose gel for analysis. If a given RNP was determined to be effective, it was used further for analysis of F0 zebrafish. As an additional confirmation of this approach, we cloned and sequenced individual target molecules to assess the percent efficacy of each gRNA used in this approach. Study-wide indel efficiency was 86%; for most gRNAs we were unable to recover non-mutant sequences from individual clones taken from injected fish, suggesting a very high degree of mutation (Figure 1).

Brightfield microscopy. Injected fish were analyzed between 1–6 dpf for potential phenotypes using an Olympus MVX-10 Macro Zoom microscope. Findings searched included: cerebral hemorrhage, aortic hemorrhage, scoliosis/axial curvature, and abnormalities of cardiac and vascular morphology (dorsal aorta (DA) and intersegmental vessels (INV)).

Confocal microscopy. In the case of *emilin1a/b* and *col5a1* transients, we analyzed *Tg(kdr:GFP)zn1* and *Tg(kdrl:ras-mCherry; fli1a:nls-GFP)s896/y7* transgenic lines using a Leica SP8 confocal microscope to identify phenotypes not visible under brightfield. Here, we assessed the diameter and endothelial cell density of the dorsal aorta (DA) and intersegmental vessels (ISVs) of the trunk.

## 3. Results

### 3.1. Zebrafish Modeling

We examined the impact of the loss of function of multiple genes on anatomy and clinical outcome in zebrafish.

Specifically, we examined two genes which have recently arisen in our clinical practice: *EMILIN1* and *MIB1.* We chose these two genes because they have presented mutations in our clinical practice in patients with thoracic aortic disease. It must be emphasized here that due to technical obstacles, we are not recreating the exact mutations in zebrafish as we observe clinically—rather, we are generating loss-of-function alleles in the associated genes. The pertinent clinical pedigrees in these families are shown in Figure 2.

The *EMILIN1* (or, elastin microfibril interface 1) gene is an extracellular matrix glycoprotein abundantly expressed in elastin-rich tissues (such as the blood vessels, skin, heart, and lung) and primarily localized at sites where elastin and microfibrils are in proximity [19]. Defects in *EMILIN1* expression adversely affect lamellar structure and function and increase aortic wall inflammation and have been associated with connective tissue disease [19,20,21,22]).

The *MIB1* gene (“Mindbomb” gene) encodes the E3 ubiquitin-protein ligase MIB1, which regulates endocytosis of Notch ligands [23] and has been associated with left ventricular non-compaction syndrome (OMIM 615092) [24].

At the earliest stages of zebrafish development (1–2 days post fertilization-dpf) the zebrafish dorsal aorta is comprised of a single-celled endothelial tube rather than the multilayered structure that is characteristic of the mature mammalian aorta [25,26]). Although zebrafish embryos will progressively add mural cells to develop a multilayered aorta during the late larval and juvenile developmental periods [25], the aorta initially resembles mammalian capillaries in structure. Consequently, attempts to model TAA in the zebrafish embryo must be both careful and broad in scope.

We were able to identify several abnormal anatomic zebrafish characteristics in our “positive controls” and in our novel variants needing evaluation for deleterious effects, namely *emilin1* and *mib1*.

These abnormal characteristics are not seen in normal zebrafish not subjected to mutagenesis of deleterious genes. As depicted in Figure 3, Figure 4 and Figure 5, these aberrant characteristics include (in various combinations for the deleterious genes):

Peri-aortic hemorrhage (manifested as red splotching adjacent to the aorta).

Cranial hemorrhage (manifested as red splotching in the brain).

Scoliosis (manifested as curvature of the spine upward toward the head, at times producing marked structural deformity of the entire body of the zebrafish).

Reduced diameter of the dorsal aorta (DA) and the intersegmental vessels (ISVx) (similar to mouse phenotype [22,23,24,25,26,27]) and “drop out” interruption of the DA.

Lower cell counts in the dorsal aorta and the intersegmental vessels

Cardiomegaly (or, more properly, distention of the pericardial sac, at times severe).

### 3.2. Genetic Modifications

Specific details for each genetic modification of the zebrafish are supplied below. The frequency of observed abnormal manifestations varied for each targeted gene, but each target displayed some degree of phenotypic abnormality (Figure 1F–G).

#### 3.2.1. Fibrillin 1 and Collagen Subunits

We initially addressed a suite of genes previously implicated in human cases and animal models of TAA [4]. These are the “positive controls” for our study, designed to show that the zebrafish model can detect adverse consequences from known pathogenic genes.

We injected CRISPR/Cas9 against *fbn1* (Marfan gene) and the collagen genes *col1a2*, *col5a1*, and *col5a2a/b* (Ehlers–Danlos genes). The most pronounced defect across all conditions was some degree of cardiac edema or cardiomegaly (Figure 3A–E, filled arrowheads). Scoliotic or kyphotic phenotypes were also frequently observed across all conditions. In a small subset of targeted fish, we observed hemorrhages of the dorsal aorta (Figure 3F, filled arrowhead)—a phenotype we never observe in control fish and which seems directly comparable to rupture or dissection of human TAA.

We proceeded to collect quantitative data on the vasculature of the trunk in *col5a1* fish. Using the *Tg(kdr:gfp)* fish line, which labels all vascular endothelial cells, we were able to visualize and measure both the dorsal aorta (DA) and the intersegmental vessels (ISVs) of the trunk (Figure 3G,H). Although we did not observe statistically significant differences in the diameter of these vessels (Figure 3I,J), we did observe missing vascular segments in both DA and ISVs as well as inappropriate vascular branching in ISVs (Figure 3H, empty arrowheads). Taken together, these data suggest that knocking out known TAA-associated genes produces abnormal vascular phenotypes in larval zebrafish as young as 3 dpf.

#### 3.2.2. Mindbomb 1

Mindbomb 1. *Mib1* is a known E3 Ubiquitin ligase that regulates Notch signaling. It is well known in zebrafish to regulate both angiogenesis and neurogenesis [28,29,30]. We designed CRISPR/Cas9 RNPs targeting the same locations as the clinically identified variants of unknown significance (Figure 4A) and then validated these using a standard T7 endonuclease assay to verify that they were generating indel variants at those locations (Figure 4B).

Injection of *mib1* CRISPR/Cas9 generated extremely severe phenotypes. Axial curvature was massively disrupted (Figure 4C,D). We also routinely observed hemorrhaging, both in the cerebral vasculature and in the aorta (Figure 4E–G). This rate of hemorrhage was significantly greater in *mib1*-deficient fish when compared to very rare events in controls (Figure 4H, chi-square test, *p* < 0.001).

#### 3.2.3. Emilin 1a and 1b

Elastin microfibril interfacer 1 (Emilin1) is an extracellular matrix protein that associates with microfibrils and elastin; it is therefore a compelling candidate for a gene implicated in TAA, which relies on the proper development and maintenance of elastic tissues such as those found in large-diameter blood vessels. We again generated CRISPR/Cas9 targeting the sites of each of the observed patient variants (Figure 5A) and validated these reagents as before (Figure 5B,C). Initially, we found no gross phenotypic abnormalities in *emilin1a/b* knockout fish (Figure 5D,E). Consequently, we proceeded to do a more careful analysis of the zebrafish trunk vasculature.

Morphological analysis of the *emilin1a/b* knockout fish revealed that both DA and ISVs exhibited significantly smaller diameters. This phenotypic profile is entirely consistent with previous reports in mouse which revealed the same findings [22,27]. We subsequently performed cell counts in the endothelial cells of these vessels using the *Tg(fli1a:nls-gfp)* line to determine whether this reduced size was due to fewer endothelial cells being present. In both vessel types, endothelial cell number was substantially reduced (Figure 5L,M). A final analysis of the DA to assess its tortuosity (Figure 5G, blue dashed lines) is suggestive of an overall trend towards greater sinuousness but ultimately was not statistically significant.

## 4. Discussion

Zebrafish have substantial features that make them very suitable for genetic investigations of the cardiovascular system and aorta. They have rapid gestation and maturation. They are transparent, so that the heart and aorta can be easily visualized by direct inspection. They can be examined by light microscopic videography. Medications can be administered by simple instillation into the bath water, from which they are systemically absorbed through the gills or skin. In short, zebrafish provide a valuable tool for study of aortic pathologies associated with specific genetic mutations.

Zebrafish have been used in non-clinical laboratory studies of various aspects of aortic disease. Abrial and colleagues at Massachusetts General Hospital have used zebrafish to study post-injury myocardial regeneration pathways as well as development of the aortic arch vessels during embryogenesis. They engineered zebrafish lacking two genes in the arch development pathway; surprisingly, those zebrafish also within days developed a Marfan’s-like phenotype, including aortic aneurysm [31]. Ton and colleagues studied a knockdown of a *Col22a1* gene in zebrafish, finding a 2.5-fold increase in the number of cerebral hemorrhages compared to the wild type zebrafish, which they attributed to abnormal vessel wall permeability [32]. They also noted associated vessel dilatation, irregular vessel contour, and irregular vascular wall structure. They found that the frequency of hemorrhages increased with heat stress (33.5 °C), forced rapid swimming, or monitoring to older zebrafish age (up to 2 years) [32]. In their studies of the toxicity of cigarette products on the cardiovascular system of zebrafish, Folkesson and colleagues found distinct evidence of toxicity to the vascular tree and aorta, manifested as cerebral hemorrhages and/or aortic dilatation [33]. Dietz and Milewicz also found zebrafish models of aortic disease useful in their basic science studies of aortic aneurysm and bicuspid valve disease [34,35]. In addition to the zebrafish aneurysm related characteristics that we observed, some authors have noted pericardial edema, maldevelopment of the aortic arches, and major developmental abnormalities of the body of the zebrafish [33].

In the present paper, we present our initial experience using zebrafish mutants with an eye toward permitting rapid, low cost, clinical estimation of the clinical significance of mutations encountered in the course of standard clinical aortic care.

We feel that this initial experience with evaluation of potential aortic aneurysm causative genetic variants by targeted genes in the zebrafish is encouraging. A zebrafish “aortic phenotype” is emerging through these initial investigations in the zebrafish mutant models. This involves hemorrhage (peri-aortic or cerebral), cardiomegaly, scoliosis, decreased vessel caliber, and decrease in number of vascular cells. This phenotype is certain to vary in zebrafish by gene, just as in humans the different genetically mediated syndromes have different manifestations.

The research reported here has several addressable limitations. First, our initial approach does not precisely duplicate the mutations observed in human patients, which are often missense mutations involving single amino acid substitutions. Rather, we generate nonsense loss-of-function mutations at the same loci as the observed patient mutation. In essence, we are really investigating the involved gene rather than the precise clinical variant itself. However, this approach is still conceptually useful as we believe many of the human mutations are indeed loss-of-function alleles based on in silico prediction; however, it is fair to say that our fish models are not one-to-one duplicates of the observed human mutations. Furthermore, if the native variant produces a nonsense or stop codon, our current work is equivalent and properly representative of the impact of the variant.

This, our initial study, may, thus, exaggerate the impact of the human mutation. To address this limitation, we will expand our study by attempting to rescue these loss-of-function alleles using injected mRNA cloned from patients carrying the exact mutation we wish to model. A functioning variant mRNA will rescue phenotypic defects. Conversely, a human mutation affecting protein functionality will be unable to rescue the zebrafish mutants.

A second limitation lies with the timing of our phenotypic characterization, which is restricted to the embryonic and larval stages. We studied only F0 zebrafish. Extending experimental time would permit assessment of F1 and F2 generations, thus minimizing the potential for persistence of *wt* fish in the F0 generation. We plan to investigate our VUS mutant phenotypes longer into the zebrafish lifecycle. This likely will permit identification of additional pathophysiological consequences beyond the TAA-related defects associated with embryonic malformation.

Additionally, the COL5A1 and COL5A2 genes chosen as positive controls represent classical Ehlers–Danlos syndrome, which is not strongly aortic but more soft tissue oriented. We plan in future experiments to test additional, more strongly aortic genes. However, even with the COL genes, we did find both aortic and skeletal abnormalities in the present studies—reflecting high discriminatory abilities of our model. As well, we did include the strongly aortic FBN1 gene even in these early studies, which proved a strong positive control.

Additionally, Marfan’s disease is known to be related, in many cases, to dominant negative activity of the mutant protein. Our modeling may not be reflective of such mechanisms. In the future the ability to investigate more VUSs in more TAA genes will build a phenotypic map of TAA associated VUSs that will enhance potential prediction of variant casualty. Furthermore, this approach may well expand our understanding of cellular processes associated with TAA-related anomalies. Furthermore, zebrafish VUS mutant models may enable screening for drugs or small molecules that can revert the course of these aberrant phenotypes, potentially contributing to future personalized therapeutics for the treatment of TAA. Even though this initial study has limitations, it does provide a novel opportunity for enhancing our biological and clinical understanding of TAA disease.

Our data suggest that the zebrafish mutagenesis model can evolve into a useful tool for rapid evaluation of novel genetic variants of unknown significance. If this eventuates, this is likely to contribute substantially to patient care. Specifically, a near immediate provisional answer as to “disease causing” versus “benign” variants may be facilitated. This would represent a considerable advance compared to the current status of waiting, possibly decades, for co-segregation of variant and phenotype to be demonstrated unequivocally in human families. The potential for lethal clinical events to occur in family members during these long “limbo” periods in the traditional clinical evaluation may be mitigated by early confirmation of disease induction in the zebrafish model. Of course, we would recommend that the zebrafish findings be considered only as one more avenue of variant evaluation—taken together with the modalities discussed in the introduction of this paper—namely, frequency of the variation in the general population, preservation in phylogeny, in silico analysis, and segregation of disease with the variant in clinical analysis of large families.

## Figures and Tables

**Figure 1 genes-13-00123-f001:**
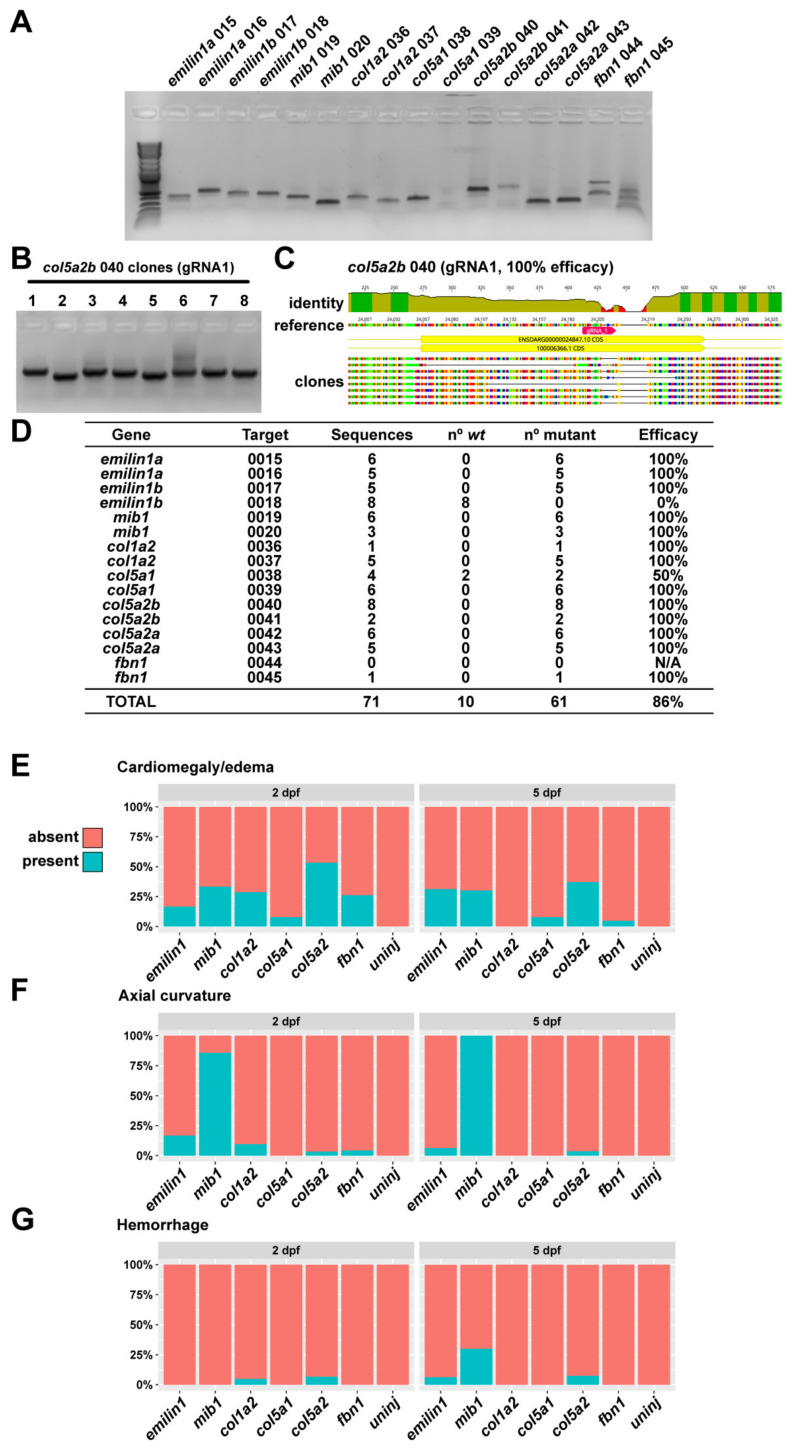
Quantification of phenotypes and gRNA efficacy. (**A**) Amplicons for each gRNA target were generated by PCR and then TOPO-cloned to generate a library of targeted genomic DNA sequences. (**B**) Individual clones were amplified by colony PCR and sent for sequencing. (**C**) Sequences were aligned to *wt* to determine the incidence of indel mutations. (**D**) For most targets, we are unable to isolate *wt* DNA by cloning, indicating a very high degree of mutational efficiency (with the exception of *emilin1b* gRNA 2, which appears to be non-functional). (**E**) Quantification of cardiomegaly phenotypes for all genes at 2 and 5 dpf. (**F**) Quantification of axial curvature defects for all genes at 2 and 5 dpf. (**G**) Quantification of hemorrhage defects for all genes at 2 and 5 dpf.

**Figure 2 genes-13-00123-f002:**
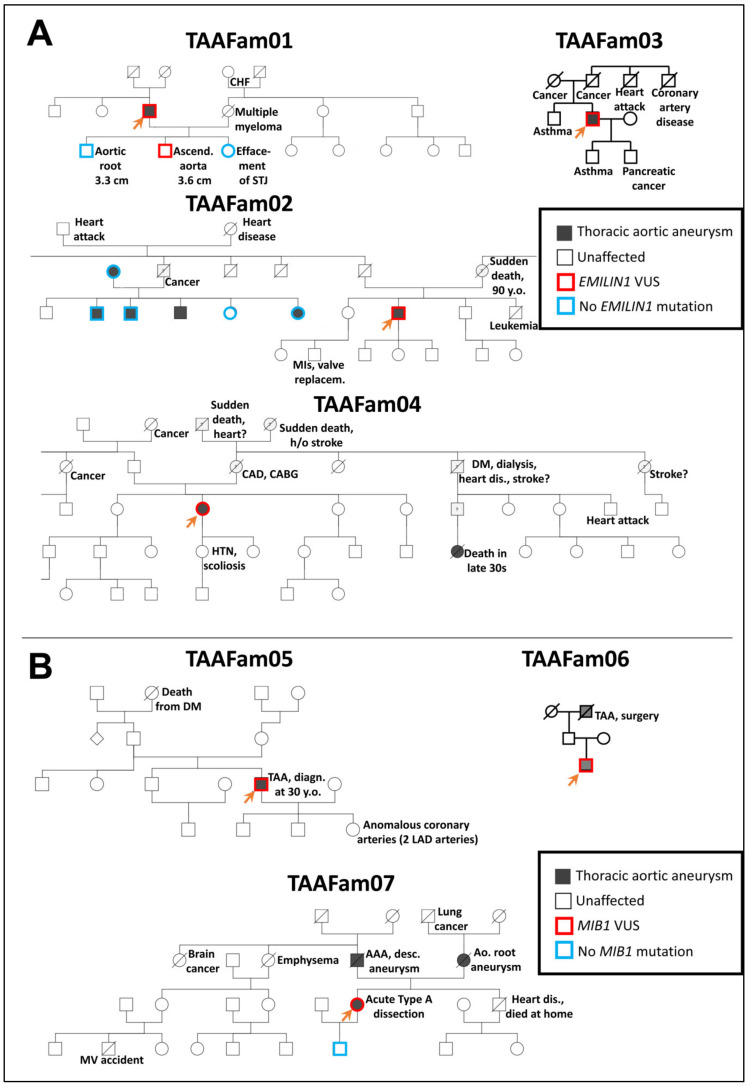
Pedigrees of *EMILIN1* (Panel **A**) and *MIB1* (Panel **B**) families with thoracic aortic disease tested as part of routine clinical genetic testing at the Yale Aortic Institute. For testing *EMILIN1* variants in the zebrafish model, missense variants identified in TAAFam02 and TAAFam03 were used (p.E170G and p.T904S, respectively). For testing *MIB1* variants in the zebrafish model, nonsense variants identified in TAAFam05, TAAFam06, TAAFam07 were used (p.R906X seen both in TAAFam05 and TAAFam06 and p. G596X for TAAFam07). The index patients (labeled with an arrow) underwent exome sequencing, while other family members underwent single-site Sanger sequencing.

**Figure 3 genes-13-00123-f003:**
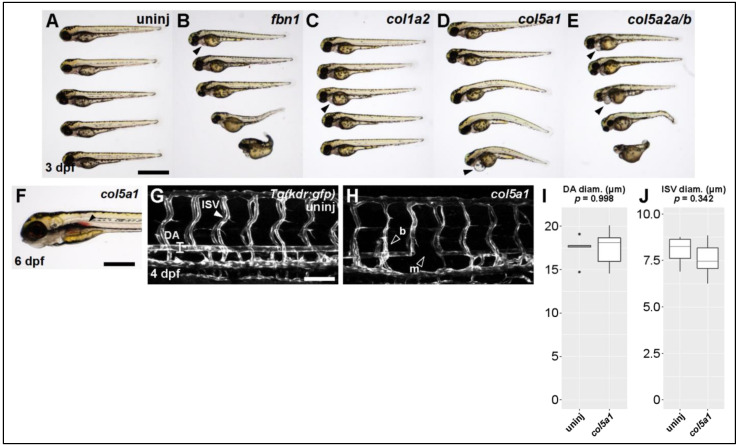
Knocking out Ehlers-Danlos and Marfan Syndrome-related genes generates vascular phenotypes. (**A**) Brightfield macroscopic image of representative series of uninjected larvae at 3 dpf. Scale bar: 1 mm. (**B**) Similar image of *fbn1* knockout fish. Filled arrowhead indicates cardiomegaly. (**C**) Similar image of *col1a2* knockout fish. (**D**) Similar image of *col5a1* knockout fish. (**E**) Similar image of *col5a2a/b* knockout fish. (**F**) Closeup brightfield image of a 6 dpf *col5a1* knockout fish with visible aortic hemorrhage (filled arrowhead, scale bar: 500 µm). (**G**) Confocal micrograph of trunk vasculature of a 4 dpf uninjected *Tg(kdr:gfp)* fish. DA: dorsal aorta, ISV: intersegmental vessel. Scale bar: 100 µm. (**H**) Similar micrograph of a *col5a1* knockout fish. Empty arrowheads indicate b: excessively branching segment and m: missing segments. (**I**) Quantification of DA diameter. Diameter is not significantly changed across conditions (*t*-test, *p* = 0.998). (**J**) Quantification of ISV diameter. Diameter is not significantly changed across conditions (*t*-test, *p* = 0.342).

**Figure 4 genes-13-00123-f004:**
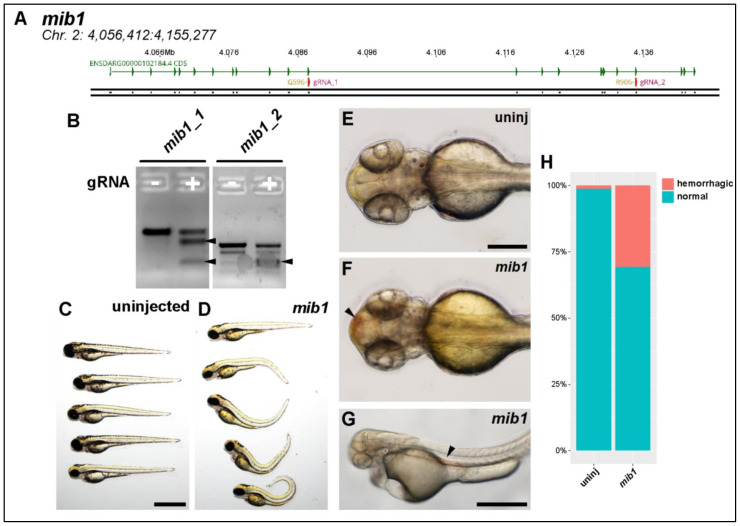
Knocking out *mib1* generates a hemorrhagic phenotype. (**A**) Genomic region of chromosome 2 containing *mib1* gene. Two patient mutations G596 and R906 are indicated. gRNAs (magenta) were designed to target both regions. (**B**) T7 endonuclease activity against gRNA target amplicons. Uninjected DNA pool (−) produces one fragmentation pattern, *mib1-*injected DNA pool (+) produces extra bands (filled arrowheads) caused by heteroduplex DNA formation due to editing and subsequent digestion by T7 endonuclease. (**C**) Brightfield macroscopic image of representative series of uninjected larvae at 3 dpf. Scale bar: 1 mm. (**D**) Similar image of *mib1* knockout fish. Note severe body curvature defects. (**E**) Uninjected 54 hpf fish, dorsal view. Scale bar: 250 µm. (**F**) Similar *mib1* knockout fish exhibiting cranial hemorrhage (filled arrowhead). (**G**) Brightfield image of 50 hpf *mib1* knockout fish with aortic hemorrhage (filled arrowhead, scale bar: 500 µm). (**H**) Quantification of hemorrhagic phenotype. *Mib1* knockout fish exhibit significantly more hemorrhagic events (uninj: 1%, *mib1*: 31%, chi-square test, *p* < 0.001).

**Figure 5 genes-13-00123-f005:**
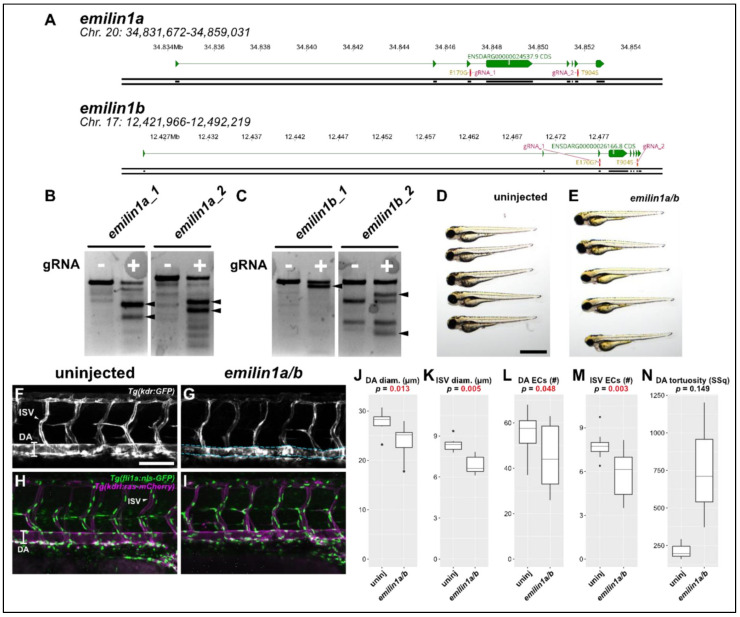
Knocking out *emilin1a/b* reduces DA and ISV diameter, endothelial cell counts. (**A**) Genomic region of chromosomes 20 and 17 containing *emilin1a* and *b*, respectively. Patient mutations E170G and T904S are indicated and their corresponding gRNAs are overlaid (magenta). (**B**) T7 endonuclease activity against gRNA target amplicons for *emilin1a*. Supernumerary bands are generated for both injected gRNAs (filled arrowheads). (**C**) Similar T7 assay as (**B**), but applied to *emilin1b* target amplicons. (**D**) Brightfield macroscopic image of a representative series of uninjected 3 dpf larvae. Scale bar: 1 mm. (**E**) Similar image of *emilin1a/b* fish. Note that fish are generally normal in appearance. (**F**) Confocal micrograph of uninjected 3 dpf *Tg(kdr:gfp)* trunk vasculature. DA: dorsal aorta, ISV: intersegmental vessel, scale bar: 100 µm. (**G**) Similar micrograph taken of an *emilin1a/b* knockout fish. DA is traced in blue to emphasize tortuosity. Note the generally lower diameter of vessels, (**H**) Uninjected 3 dpf *Tg(fli1a:nls-gfp); Tg(kdrl:ras-mcherry)* trunk vasculature. DA: dorsal aorta, ISV: intersegmental vessel. Endothelial cell nuclei are labelled with GFP, making quantification possible. (**I**) Similar micrograph taken from an *emilin1a/b* knockout fish. These parameters are qualified in the following five panels: (**J**) DA diameter, (**K**) ISV diameter, (**L**) DA endothelial cell count, (**M**) ISV endothelial cell count, (**N**) DA tortuosity. With the exception of (**N**) all parameters are significant as indicated (*t*-test).

**Table 1 genes-13-00123-t001:** scRNAs and primers used to perform T7 assays. For each gene, two target-specific scRNAs were designed. Flanking primers generating amplicons between 300–600 bp were also generated to assess whether indel mutations were being generated at the desired locations.

Target Gene	scRNA Sequence		Forward Primer		Reverse Primer
*col1a2*	CCTGGCCCCCCTGGTCTTGG	5′	CCAGTTGTTTGATTATATTGTTCTG	5′	GTCTCAAATATAACAACAAACACTC
*col1a2*	CAGCACCGCGAGCACCAGCT	5′	CTACTTATTTGAACTGTTCACTTTG	5′	TAATCTTCTAGCCACATTATTTCAC
*col5a1*	CCGAAACGGAACTTCTGGAT	5′	GATATGAAACCTATTATGACGAGTC	5′	TAGATAACTAAACTCACTGAGGTAG
*col5a1*	AGCATCGGTGTAATCCGTGA	5′	GAGAAGGAGATTATACTGTAGGAG	5′	CTGTTATAAATGAACAAGTCATTCC
*col5a2a*	TGCCAGGGCGATGCAGCTGA	5′	GATGAGTTGAGCTGTACAGAG	5′	AAGAAGAAACATAACAGACAAAAAC
*col5a2a*	AGGCACTAATGCCCGGTGTT	5′	GCAGATCCAATAAAGTGATTATTTC	5′	TGTGGACAAATTATTCGTTTTTATC
*col5a2b*	CGAGCAATTCTTGATCCCGG	5′	GCAATTGAAACAAACACAATATAAC	5′	GTATTTAACGTGACGTTAAGTTAAC
*col5a2b*	CCTATAGGACCAGGCTCACC	5′	TTCTTATTTAGATTGTGGGAATACC	5′	GCACATTTTATTACTCTAATCACTG
*emilin1a*	GCAAGACCTACTGGTGAAGA	5′	TGATTAATTGTTATGTTGATGCATG	5′	AAAGTGTTACCAAATAAACAAACTC
*emilin1a*	AAGGCCTACAATCCAAAAAC	5′	CAGTAATCTAACTAATACAACTCCC	5′	ATAACATTGTTAGCATTTAACACTG
*emilin1b*	CAGGCAGACCTGGGAATGGT	5′	TTTGAAGTGAAAGTTCTGTTTAATG	5′	TGCTAAGATGATATATTCAGTGTTG
*emilin1b*	AACTTTTACAATTCACGAAC	5′	TTTGTATGAGAGTCAGTGATATTTC	5′	CTAAAGAAGTACTTTCCTTCTACTG
*fbn1*	ATACCGCTGCCTCTGCAAAG	5′	CTCATAACTGTGATCATCATGC	5′	CTCTCACACAGATTCACATTATC
*fbn1*	GAGCTCAATCCCACTGGAGT	5′	ATACCCATGTCATTATACAGATTTG	5′	CCAAATTATCCAGCTTATGTTTTAC
*mib1*	GTCACAATCACCAACAACAA	5′	TCTCTTTCTTGATTTTTGAAATTGG	5′	TACAGAAGGATTGAATAATTCTGAC
*mib1*	TGTGCAGTGTCGGGCCGTGG	5′	CAAAATTAAATGTCATACAGTGAGG	5′	AAATATTGAAGATGAGAGACATGAG

## Data Availability

Available on request.

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
