# Peer review of "Phenotyping Zebrafish Mutant Models to Assess Candidate Genes Associated with Aortic Aneurysm"

_genes, 2022, doi:10.3390/genes13010123_

Round 1

Reviewer 1 Report

I must appreciate authors enthusiasm and effort to address ‘Variants of Uncertain Significance’ (VUS) issue related to thoracic aortic aneurysm (TAA). Zebrafish is probably the best animal model system to test many of the VUS identified in clinical genetic testing. However, the methodology to study VUS in zebrafish for various genetic diseases still needs to be developed. Following are my comments or suggestions for the authors. 

  1. Authors refer multiple times in the paper that they have identified/ examined two ‘Variants of Uncertain Significance’ (VUS): EMILIN1 and MIB1. It is wrong to refer to a gene as VUS. It should be referred for variant only. VUS can be identified in a well-known disease-causing gene. For example, about 1500 VUS are identified in FBN1 gene (Clinvar). Change the text accordingly.
  2. Authors are not testing the exact VUS identified in the patients. So, it is misleading to claim that they are testing VUS variants identified in patients. This is especially more applicable for missense variants identified in EMILIN1. Instead of claiming that they are testing VUS, they should consider changing the text to testing two TAA causing candidate genes for loss-of-function mechanism.
  3. A CRISPR/Cas9 injection in F0 fish result in varying percentage of mutagenesis and different types of indel variants from one injection to another. In fact, mutagenesis is never 100%, as observed for mib1 (Figure 4B), emilin1a (Figure 5B) and emilin1b (Figure 5C). A band corresponding to uninjected control is observed for all of them. Additionally, there is 1/3rd chance that a resulting indel variant is a multiple of 3 bp, potentially going back in-frame with protein coding. So, interpretation of data directly collected from F0 fish can be misleading and/or inaccurate. Also, there is high possibility for non-specific cleavage by CRISPR/Cas9, which is not being tested here. So, characterizing embryos generated from two independent indel variant carrying F1 fish is the most common practice by many research labs. Why did authors not consider this methodology for their study?
  4. Line 216: what does ‘[19=22]’ refers to?
  5. Genes COL5A1 and COL5A2 that are used as positive control. These two genes are known to cause Classic Type Ehlers-Danlos Syndrome. In these patients, TAA is not a common phenotype. What is reason for selecting these genes when they could have selected other genes that have predominantly TAA phenotype?
  6. The most common disease mechanism described for both Marfan Syndrome and Ehlers-Danlos Syndrome is by dominant negative activity of mutant protein. What is reason for authors to choose loss-of-function methodology to study all seven genes?
  7. The fonts in Figure 1 are too small to read.
  8. Authors mention that they TOPO cloned targeted PCR product after the mutagenesis as shown in Figure 1A. They only show data for one clone for each injection. Did they test sequence multiple clones? Also, they should have run these PCR products along with uninjected control to show the difference in size.

Author Response

We thank the Reviewer for the very insightful comments and suggestions. We have implemented every one of these, as indicated in the specific responses below. This input from the Reviewer has allowed us to clarify and strengthen our manuscript considerably.

1. Authors refer multiple times in the paper that they have identified/ examined two ‘Variants of Uncertain Significance’ (VUS): EMILIN1 and MIB1. It is wrong to refer to a gene as VUS. It should be referred for variant only. VUS can be identified in a well-known disease-causing gene. For example, about 1500 VUS are identified in FBN1 gene (Clinvar). Change the text accordingly.

Thank you. We have now avoided referring to a gene as a VUS, as in the corrections implemented on lines 205-206 and 364.

2. Authors are not testing the exact VUS identified in the patients. So, it is misleading to claim that they are testing VUS variants identified in patients. This is especially more applicable for missense variants identified in EMILIN1. Instead of claiming that they are testing VUS, they should consider changing the text to testing two TAA causing candidate genes for loss-of-function mechanism.

We have expanded our disclaimer on this point by adding the Reviewer's exact terminology to the section on this important limitation. The added Reviewer's terminology is bolded in the excerpt below (lines 465-472) l. We have indicated, accordingly, that our study may exaggerate the actual impact of the human variant.

The research reported here has several addressable limitations. First, our initial approach does not precisely duplicate the VUS observed in human patients, which are often missense mutations involving single amino acid substitituons. Rather, we generate nonsense loss-of-function mutations at the same loci as the observed human mutations. This is conceptually useful as we believe many of the human VUS are indeed loss-of-function alleles based on in silico modeling, but it is fair to say that our fish models are not one-to-one duplicates of the observed human mutations. Rather, we are testing two candidate genes for loss-of-function mechanism. This, our initial study may, thus, exaggerate the impact of the human mutation. 

3. A CRISPR/Cas9 injection in F0 fish result in varying percentage of mutagenesis and different types of indel variants from one injection to another. In fact, mutagenesis is never 100%, as observed for mib1 (Figure 4B), emilin1a (Figure 5B) and emilin1b (Figure 5C). A band corresponding to uninjected control is observed for all of them. Additionally, there is 1/3rd chance that a resulting indel variant is a multiple of 3 bp, potentially going back in-frame with protein coding. So, interpretation of data directly collected from F0 fish can be misleading and/or inaccurate. Also, there is high possibility for non-specific cleavage by CRISPR/Cas9, which is not being tested here. So, characterizing embryos generated from two independent indel variant carrying F1 fish is the most common practice by many research labs. Why did authors not consider this methodology for their study?

We appreciate the reviewer’s focus on this issue, which we agree is of extreme importance. The reviewer is absolutely correct that when using a single gRNA, 1/3 of edits will not lead to a nonsense generating indel, and that F0 embryos will undoubtedly be to some extent mosaic. We address this in two ways—

We use two gRNAs per gene throughout this study. As the number of gRNAs injected per gene increases, the probability of obtaining a frameshift mutation also increases. For a gRNA that is 100% effective, injecting two gRNAs leads to a knockout probability of approximately 75% (see Kroll et al. 2021, Figure 1B for a theoretical model of this phenomenon). We could in theory drive this probability even higher by injecting additional gRNAs and in future iterations of this approach we may do that—the only real limit here is cost.

Here, our T7 assay is mostly used as an initial screening step to assess whether the gRNA works at all. The reviewer correctly notes here that the “wt” band remains in injected fish. That said, the enzyme cleaves target DNA only in the case of mismatch. That is to say, mutant DNAs carrying the same lesion that re-anneal will not cleave, and will appear identical to wt DNA. We view T7 assays as fairly crude ways of assessing gRNA efficacy, and we feel it is the general consensus of the field that even the best gRNAs never achieve total erasure of the “wt”/non-cleaved molecular species. Given the limitations of this assay, we further isolated TOPO clones from injected embryos to more directly assess the rate of gRNA efficacy. In these experiments, we generally had difficulty isolating wt clones from injected embryos, indicating that almost all our gRNAs had very high rates of efficacy (Supplemental Figure S1).

The reviewer is also correct to note that performing experiments in F1 or F2 generations would wholly eliminate the above concerns. However, doing so requires a wait time of ≥ 3 months to achieve an adult founder generation, plus additional time for screening and sequencing, plus another ≥ 3 months to achieve an adult F1 generation, plus additional time for screening and sequencing, as well as all the associated costs for per diem tank maintenance. Although the experimental outcomes would be superior, they would come at the expense of the ability to screen multiple genes rapidly and at lower expense. We are not the only group advocating for the utility of F0 screening—please see the Kroll et al. 2021 eLife study for another group that has independently arrived at this method.

We have added this important point made by the Reviewer to the limitations section of our Discussion (lines    ). We now state (lines 478-479): We studied only F0 zebrafish. Extending experimental time would permit assessment of Fa and F2 generations, thus minimizing the potential for persistence of wt fish in the F0 generation.

We emphasize throughout that ours is a preliminary investigation suggesting zebrafish as a viable mechanism for rapid assessment--in response to a real clinical quandry regarding interpretation of VUSs in the clinic and in patient management.

4. Line 216: what does ‘[19=22]’ refers to?

Thank you for the keen eye. That was a reference citation, which we have corrected.

5. Genes COL5A1 and COL5A2 that are used as positive control. These two genes are known to cause Classic Type Ehlers-Danlos Syndrome. In these patients, TAA is not a common phenotype. What is reason for selecting these genes when they could have selected other genes that have predominantly TAA phenotype?

We accept fully the Reviewer’s point that a more “aortic” phenotype than classic Ehlers-Danlos (cE-D) would have made a better choice for our positive control. Accordingly, in our next round of experiments, we will make it a priority to include other typically aortic variants as positive controls. We take solace in the fact that our zebrafish model did reveal substantial phenotypic aberrations even when our cE-D variant was introduced—supporting the discriminant abilities of the model.  Also, even finding extra-aortic abnormalities (especially skeletal) in the zebrafish model confirms variant pathogenicity of one type or another. Skeletal abnormalities certainly are front and center in many heritable aortic diseases. Additionally, even in the cE-D zebrafish, we did find aortic hemorrhages and missing segmental branches of the dorsal aorta (Figure 3). Also, very recent data finds that fully half of “classic” E-D human patients do show aortic enlargement. (REF: Rauser-Foltz K, Starr LJ, Yetman AT. Utilization of echocardiography in Ehlers-Danlos syndrome. Congenit Heart Dis. 2019;14:864-867. doi: 10.1111/chd.12824.) Nonetheless, we look forward to future experiments in the exact direction toward which the Reviewer encourages to explore. We have added lines of text reflecting your concerns on the COL genes to the limitations section of the Discussion (Lines 478-479).

Also, our experiments did include an FBN1 positive control, representing Mafan syndrome, which is strongly aortic in character.

6. The most common disease mechanism described for both Marfan Syndrome and Ehlers-Danlos Syndrome is by dominant negative activity of mutant protein. What is reason for authors to choose loss-of-function methodology to study all seven genes?

We thank the reviewer for pointing out that dominant negative activity is the molecular mechanism by which Marfan Syndrome and EDS are generally precipitated. Ideally, we should model some VUSs as dominant-negative mRNAs or something similar. We chose our approach for two reasons—

Dominant negative activity is not the only mechanism for all disease-causing MFS variants. Our in silico modeling for variants we have encountered in emilin1 and mib1 suggests that they are indeed loss-of-function alleles, and we anticipate that other variants may be as well.

Dominant negative mutations, although mechanistically distinct from loss-of-function mutations, may be partially replicated by loss-of-function mutations in this context. For example, FBN1 mutations leading to Marfan Syndrome are diverse, including not only missense mutations, but also premature terminations, splice mutations, and exon-skipping mutations. These generally interfere with microfibril assembly in a dominant-negative fashion—but a similar failure or deficiency in microfibril assembly would likely also be achieved by the types of mutations we induce in FBN1 by generating small indels leading to nonsense alleles. This distinction would be extremely important as to the heritability of these phenotypes, but since we are working exclusively in F0 fish for the purposes of screening, we are not seeking to model the heritability aspects of these disorders.

We have added the following statement, based on the Reviewer's comment, to our limitations section: Marfan's disease is known to be related, in many cases, to dominant negative activity of the mutant protein. Our modeling may not be reflective of such mechanisms. (Lines 478-479.)

7. The fonts in Figure 1 are too small to read.

Yes, we apologize for this. The writing within each pedigree is too small to read. This is a byproduct of the multiple pedigrees that we have included in one single figure. We can confirm, however, that, as Genes is an electronic journal, all the fonts can be read by using the "View" enlargement option within the Figure. We hope this suffices, as we felt to present all the pedigress in individual figures would be excessive, as we simply aimed to show the overall inheritance patterns.

8. Authors mention that they TOPO cloned targeted PCR product after the mutagenesis as shown in Figure 1A. They only show data for one clone for each injection. Did they test sequence multiple clones? Also, they should have run these PCR products along with uninjected control to show the difference in size.

We thank the reviewer for their attention to this important validation of gRNA efficacy, and we regret being unclear in our explanation of these results. In panel A, we are indeed showing a single gel lane for each gRNA target. However, in panel B, we attempt to show that 8 different clones (of slightly varying lengths) were isolated from col5a2b target 040, and that after sequencing each of these clones, all of them were mutated relative to wt (panel C). We have perhaps been insufficiently clear in expressing that we extended the same approach to all targets, sequencing multiple clones from each target to develop a gRNA efficacy table for every gRNA used in the study (panel D). Although we were not always able to isolate 8 clones per target, we isolated almost exclusively modified DNA when using this approach. We have attempted to clarify this in the results section.

Reviewer 2 Report

The authors presented the paper describing:  Phenotyping Zebrafish Mutant Models to Assess Aortic Aneurysm Genetic “Variants of Uncertain Significance”.
The problem of determining the causality of VUS in the genetic diagnosis of monogenic diseases, including familial forms of thoracic aortic aneurysm, is undoubtedly serious. Nevertheless, the experimental data presented in the work show only the possibility of confirming the relationship of genes with the development of TAA, using Zebrafish Mutant Models, but not specific genetic variants. The possibility of assessing VUS is no more than an assumption.
Thus, the article must be corrected, including the title, and the conclusion. The main focus should be on assessing the relationship of genes to TAA using the Zebrafish model. 
The introduction section, lines 76-113, may be shortened, as this is well-known information from the ACMG 2015 guidelines

Author Response

The authors presented the paper describing:  Phenotyping Zebrafish Mutant Models to Assess Aortic Aneurysm Genetic “Variants of Uncertain Significance”.

The introduction section, lines 76-113, may be shortened, as this is well-known information from the ACMG 2015 guidelines.

Dear Reviewer, we have cut two dispensable sentences from the Introduction. We beg your indulgence with the remainder of this section for the following reason: We hope that this manuscript will prove to be of value to audiences in addition to experts in genetics and molecular genetics. Specifically, we hope that this paper will be of use to cardiologists and cardiac surgeons, as well as vascular surgeons. Those groups will not be familiar with the concepts in our Introduction, which, as you point out, are well-known to the usual Genes audience. We kindly request your permission to preserve the remainder of the Introduction.